# A CLOSER LOOK AT NETWORK RESOLUTION FOR EFFICIENT NETWORK DESIGN

## ABSTRACT

There is growing interest in designing lightweight neural networks for mobile and embedded vision applications. Previous works typically reduce computations from the structure level. For example, group convolution based methods reduce computations by factorizing a vanilla convolution into depth-wise and point-wise convolutions. Pruning based methods prune redundant connections in the network structure. In this paper, we explore the importance of network input for achieving optimal accuracy-efficiency trade-off. Reducing input scale is a simple yet effective way to reduce computational cost. It does not require careful network module design, specific hardware optimization and network retraining after pruning. Moreover, different input scales contain different representations to learn. *We propose a framework to mutually learn from different input resolutions and network widths*[1]. With the shared knowledge, our framework is able to find better width-resolution balance and capture multi-scale representations. It achieves consistently better ImageNet top-1 accuracy over US-Net (Yu & Huang, 2019) under different computation constraints, and outperforms the best compound scale model of EfficientNet (Tan & Le, 2019) by 1.5%. The superiority of our framework is also validated on COCO object detection and instance segmentation as well as transfer learning.

## 1 INTRODUCTION

Deep neural networks have triumphed over various perception tasks such as image classification (He et al., 2016; Huang et al., 2017b; Simonyan & Zisserman, 2014), object detection (Ren et al., 2015; Redmon et al., 2016) and semantic segmentation (Chen et al., 2017). However, deep networks usually require large computational resources, making them hard to deploy on mobile devices and embedded systems. This motivates research in reducing the redundancy in deep neural networks or designing light-weight structures. Specifically, MobileNet (Howard et al., 2017) factorizes a standard $3 \times 3$ convolution into a $3\times3$ depth-wise convolution and a $1\times1$ point-wise convolution. ShuffleNet (Zhang et al., 2018a) uses $1 \times 1$ group convolution to further reduce computations and proposes the shuffle operation to help information flow among different groups. Another kind of approach is to prune redundant connections in the networks. Unstructured pruning methods (Han et al., 2015b;a) delete network connec-

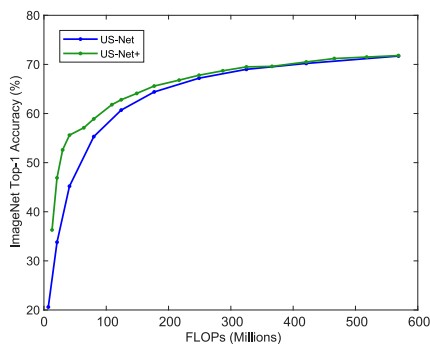

Figure 1: Accuracy-FLOPs curve of US-Net+ and US-Net. US-Net+ means simply applying different resolutions to US-Net during testing.

tions which are thought unimportant. Structured pruning methods (Li et al., 2016; Anwar et al., 2017; Lemaire et al., 2018) prune the entire filters and feature maps. Recently, dynamic networks have been introduced which adopt a single model to meet varying computing resource constraints. For example, Huang et al. (2017a) proposes a multi-branch structure where early prediction can be made based on the current confidence and resource constraints. Kim et al. (2018) builds a network-

---

[1]Number of channels in a layer.

in-network structure for multiple resources. Yu et al. (2019) shares weights among different sub-networks and each sub-network has its own batch normalization layer. US-Net (Yu & Huang, 2019) proposes to compute batch normalization statistics after training, and introduces two training techniques to train a network that is executable at any network widths. However, these works approach the problem only from network structure perspective, ignoring the importance of network input.

Reducing network input dimension (e.g., lowering image resolution) is a straightforward way to reduce computational cost. It can be applied to any network structure during testing. Besides, balancing between input resolution and network width can achieve better accuracy-efficiency tradeoffs. For example, to meet a dynamic resource constraint from 13 MFLOPs to 569 MFLOPs on MobileNet v1 backbone, US-Net (Yu & Huang, 2019) needs a network width range of $[0.05\times, 1.0\times]$ given a $224\times224$ input resolution. This constraint can also be met via a network width of $[0.25\times, 1.0\times]$ by adjusting the input resolution from $\{224, 192, 160, 128\}$ during test time. We denote the latter model as US-Net+. As shown in Figure 1, simply combining different resolutions with network widths during testing can already achieve a better accuracy-efficiency tradeoff than US-Net without additional efforts. EfficientNet (Tan & Le, 2019) also acknowledges the importance of balancing between depth, width and resolution. Moreover, reducing input resolution does not necessarily harm the performance, and may sometimes even be beneficial. Chin et al. (2019) points out that lower image resolution may produce better detection accuracy by reducing focus on redundant details. Chen et al. (2019b) claims that different scaled images contain different information. Sun et al. (2019) fuses multi-scale features with a multi-branch framework to learn robust representations.

Inspired by the observations above, we propose a *unified framework to mutually learn from input resolution and network width*. Our framework is able to achieve the optimal width-resolution balance under certain resource constraint. Since we share weights among different network widths, each network is able to capture multi-scale representations without any adjustments in the network structure. The whole framework is shown in Figure 2. We summarize our contributions as follows.

1. We highlight the importance of input resolution for efficient network design. Previous works either ignore it or treat it independently from network structure. In contrast, we propose a unified framework to mutually learn from input resolution and network width.

2. Our framework is simple and general. It is compatible with any network structures (i.e., network-agnostic) and is as simple as training an independent network. Moreover, it does not change network structure, meaning it can benefit from other add-on structural optimizations.

3. We conduct extensive experiments to verify the effectiveness of our framework. Our framework significantly outperforms US-MobileNets (Yu & Huang, 2019) and independently-trained MobileNets on ImageNet classification, as well as COCO object detection and instance segmentation. The experimental results on popular transfer learning datasets also demonstrate the generalization ability of the learned representations of our model.

## 2 RETHINKING REDUCING COMPUTATIONAL COST

In this section, we explore the importance of input resolution for efficient network design. The computational cost of a vanilla convolution is

$$C_1 \times C_2 \times K \times K \times H \times W \qquad (1)$$

where $C_1$ and $C_2$ are the number of input and output channels, $K$ is kernel size, $H$ and $W$ are output feature map size. Most previous works only focus on reducing computational cost from the structure level, that is, reducing the number of channels $C_1 \times C_2$ or reducing kernel size $K$. MobileNets (Sandler et al., 2018; Howard et al., 2017) decompose the vanilla convolution into a depth-wise convolution and a $1\times1$ convolution, reducing the cost to $(C_1 \times K \times K + C_1 \times C_2) \times H \times W$. ShuffleNet (Zhang et al., 2018a) further divides $1 \times 1$ convolution into several groups, shrinking the computational cost to $(C_1 \times K \times K + C_1 \times (C_2/g)) \times H \times W$, where $g$ is the number of groups. However, increasing the number of groups leads to high memory access cost (MAC) (Ma et al., 2018), which makes the network inefficient in practical applications. Pruning based methods aim to prune redundant connections in the network, cutting the computational cost to $\beta_1 \times C_1 \times \beta_2 \times C_2 \times K \times K \times H \times W$, where $\beta_1$ and $\beta_2$ are pruning ratios. Unstructured pruning methods (Han et al., 2015b;a) remove the network connections which are thought unimportant. The outcome is a sparsely connected network, which is not efficient on standard libraries and needs

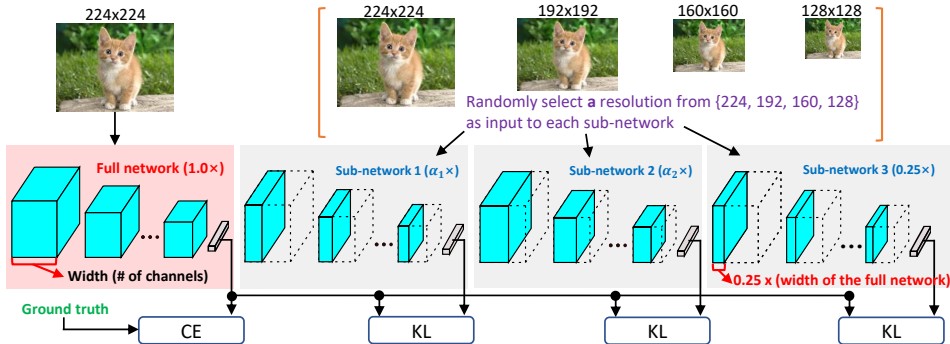

Figure 2: The training process of our proposed framework. The network width range is [0.25, 1.0], input resolution is chosen from {224, 192, 160, 128}. This can achieve a computation range of [13, 569] MFLOPs on MobileNet v1 backbone. We follow the *sandwich rule* (Yu & Huang, 2019) to sample 4 networks, i.e., upper-bound full width network ($1.0\times$), lower-bound width network ($0.25\times$), and two random width ratios $\alpha_1, \alpha_2 \in (0.25, 1)$. For the full-network, we constantly choose 224×224 resolution. For the other three sub-networks, we randomly select its input resolution. The full-network is optimized with the ground-truth label. Sub-networks are optimized with the prediction of the full-network. Weights are shared among different networks to facilitate mutual learning. CE: Cross Entropy loss. KL: KL Divergence loss.

specialized software and hardware optimization (Han et al., 2016). Structured pruning methods (Li et al., 2016; Anwar et al., 2017; Lemaire et al., 2018) prune the entire filters and feature maps. The pruned network maintains structured but needs retraining to retain performance after pruning, which makes it hard to meet the dynamic resource constraints in real-world applications. US-Net (Yu & Huang, 2019) addresses the dynamic constraint problem by sharing weights among different sub-networks and optimizing them simultaneously. But it only considers reducing network width and the performance drops dramatically as computational resource goes down.

Instead of only focusing on $C_1 \times C_2$, we shift our attention to reducing $H \times W$ in Eq. 1, i.e., lowering input resolution for the following three reasons. First, reducing $H$ and $W$ can reduce the computational cost without making any adjustments to the network structure. Therefore, it does not require further hardware optimization (Han et al., 2016) or careful structure tuning (Ma et al., 2018). Second, balancing between input resolution and network width will produce better accuracy-efficiency tradeoffs as shown in Figure 1. Third, different resolutions contain different information. Lower resolution images may contain more global structures while higher resolution ones may encapsulate more fine-grained patterns. Several works (Chen et al., 2019b; Sun et al., 2019; Ke et al., 2017) have explored such multi-scale representation learning. However, they resort to a multi-path structure, which is unfriendly to parallelization (Ma et al., 2018).

Motivated by the discussion above, we propose a unified framework to mutually learn from network width and input resolution. Our framework is able to find the optimal width-resolution configuration under certain constraint, and captures multi-scale representations by sharing weights. In addition, it is compatible with any network structures and does not need further optimization and retraining.

## 3 METHODOLOGY

### 3.1 PRELIMINARY

Our framework leverages the training techniques in US-Net (Yu & Huang, 2019). Therefore, we first introduce these techniques in this section to make this paper self-contained.

**Sandwich Rule.** US-Net trains a network that is executable at any resource constraint. The solution is to randomly sample several network widths for training and accumulate their gradients for optimization. However, the performance of all the sub-networks is bounded by the smallest width (e.g., $0.25\times$) and the largest width (e.g., $1.0\times$). Thus, the authors introduce the *sandwich rule* that is always sampling the smallest and largest width plus two random widths for each training iteration.

**Inplace Distillation.** Knowledge distillation (Hinton et al., 2015) is an effective method to transfer knowledge from a teacher network to a student network. Following the *sandwich rule*, since the largest network is sampled in each iteration, it is natural to use the largest network as the teacher to guide smaller sub-networks. The largest network is supervised by the ground truth label. This gives a better performance than only training with ground truth labels.

**Post-Statistics of Batch Normalization.** US-Net proposes that each sub-network needs their own batch normalization (BN) statistics (mean and variance), but it is insufficient to store the statistics of all the sub-networks. As a result, US-Net collects BN statistics for the desired sub-network after training. Experimental results show that 2,000 samples are sufficient to get accurate BN statistics, so this procedure is very efficient.

## 3.2 MUTUAL LEARNING FROM WIDTH AND RESOLUTION

As discussed in Section 2, different resolutions contain different information. We want to take full advantage of this attribute to learn robust representations. In US-Net, *sandwich rule* is proposed to train a network that is executable at any width. However, we view it as a scheme of mutual learning (Zhang et al., 2018b). Since networks with different widths share weights with each other, larger networks can take advantage of the features captured by smaller networks. Also, smaller networks can benefit from the stronger representation ability of larger networks. In light of this, we propose to mutually learn from network widths and resolutions.

**Training framework.** Our proposed framework is presented in Figure 2. We train a network where its width ranges from $0.25\times$ to $1.0\times$. We first follow the *sandwich rule* to sample four sub-networks, i.e., the smallest ($0.25\times$), the largest ($1.0\times$) and two random ones. Then, unlike traditional ImageNet training with $224\times224$ input, we resize the input image to four resolutions $\{224, 196, 160, 128\}$ and feed them into different sub-networks. We denote the weights of a sub-network as $W_{0:w}$, where $w$ is the width of the sub-network and $0:w$ means the sub-network adopts the first $w \times 100\%$ weights of each layer of the full network. $I_{R=r}$ represents a $r \times r$ input image. Then $N(W_{0:w}, I_{R=r})$ represents a sub-network with width $w$ and input resolution $r \times r$. For the

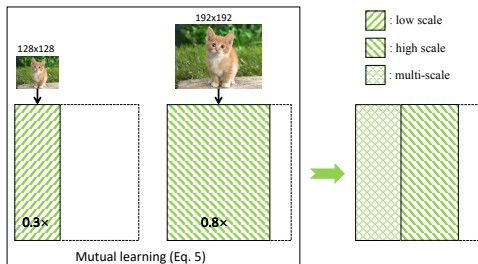

Figure 3: Illustration of the mutual learning from width and resolution scheme.

largest sub-network (i.e., the full-network in Figure 2), we always train it with the highest resolution ($224 \times 224$) and ground truth label $y$. The loss for the full network is

$$loss_{full} = CrossEntropy(N(W_{0:1}, I_{R=224}), y). \tag{2}$$

For the other sub-networks, we randomly pick an input resolution from $\{224, 196, 160, 128\}$ and train it with the output of the full-network. The loss for the $i$-th sub-network is

$$loss_{sub_i} = KLDiv(N(W_{0:w_i}, I_{R=r_i}), N(W_{0:1}, I_{R=224})), \tag{3}$$

where $KLDiv$ is the Kullback-Leibler divergence. The total loss is the summation of the full-network and sub-networks, i.e.,

$$loss = loss_{full} + \sum_{i=1}^{3} loss_{sub_i}. \tag{4}$$

The reason for training the full-network with the highest resolution is that the highest resolution contains more details. Also, the full-network has the strongest learning ability and captures the detailed dicriminatory information from the image data. We experiment with randomly selecting resolutions for all four widths, but it yields worse performance.

**How mutual learning works.** In this part, we are going to explain why the proposed framework can mutually learn from different widths and resolutions. For simplicity, we only consider two network widths $0.3\times$ and $0.8\times$, and two resolutions 128 and 192 in this example. As shown in Figure 3, sub-network $0.3\times$ selects input resolution 128, sub-network $0.8\times$ selects input resolution 192. Then we can define the gradients for sub-network $0.3\times$ and $0.8\times$ as $grad_{W_{0:0.3}, I_{R=128}}$ and

$grad_{W_{0:0.8}, I_{R=192}}$, respectively. Since $0.8\times$ shares weights with $0.3\times$, we can decompose its gradient as $grad_{W_{0:0.8}, I_{R=192}} = grad_{W_{0:0.3}, I_{R=192}} + grad_{W_{0.3:0.8}, I_{R=192}}$. The gradients of the two networks are accumulated during training, and the total gradients are computed as

$$
\begin{aligned}
grad &= grad_{W_{0:0.3}, I_{R=128}} + grad_{W_{0:0.8}, I_{R=192}} \\
&= (grad_{W_{0:0.3}, I_{R=128}} + grad_{W_{0:0.3}, I_{R=192}}) + grad_{W_{0.3:0.8}, I_{R=192}}.
\end{aligned}
\tag{5}
$$

Therefore, the gradient for sub-network $0.3\times$ is $(grad_{W_{0:0.3}, I_{R=128}} + grad_{W_{0:0.3}, I_{R=192}})$, which enables it to capture multi-scale representations from different input resolutions. Thanks to the random sampling of network width, every sub-network is able to learn multi-scale representations in our framework.

**Model Inference.** The trained model is executable at various width-resolution configurations. The goal is to find the best configuration under a particular resource constraint. A simple way to achieve this is via query table. Specifically, we sample network width from $0.25\times$ to $1.0\times$ with a step-size of $0.05\times$, and sample network resolution from $\{224, 192, 160, 128\}$. We test all these width-resolution configurations on a validation set and choose the best one under a given constraint (FLOPs or latency). Since there is no retraining, the whole process is once for all.

## 4 EXPERIMENTS

In this section, we first present our results on ImageNet classification (Deng et al., 2009). We conduct extensive experiments to illustrate the effectiveness of the proposed framework. Next, we fine-tune the pre-trained model on popular transfer learning datasets to demonstrate the robustness and generalization ability of the learned representations. We further apply our framework to COCO object detection and instance segmentation. **To the best of our knowledge, we are the first to benchmark arbitrary-constraint dynamic networks on detection and instance segmentation.**

### 4.1 IMAGENET CLASSIFICATION

We compare our framework with US-Net and independently-trained networks on the ImageNet dataset. We evaluated our framework on two popular light-weight structures, MobileNet v1 (Howard et al., 2017) and MobileNet v2 (Sandler et al., 2018). These two networks also represent non-residual and residual structures respectively. We follow the training setting in US-Net (Yu & Huang, 2019). Please refer to the details in Appendix A.1.

**Compare with US-Net.** We first compare our framework with US-Net on MobileNet v1 and MobileNet v2 backbones. The Accuracy-FLOPs curve is shown in Figure 4. We can see that our framework consistently outperforms US-Net on both MobileNet v1 and MobileNet v2 backbones. Specifically, we achieve significant improvement under small computational cost. This is because our framework can find a better balance between network width and resolution. For example, if the resource constraint is 150 MFLOPs, US-Net reduces the width to $0.5\times$ but our framework selects a balanced configuration of $0.7\times$ - 160. Moreover, our framework even improves on the full configuration ($1.0\times$ - 224). This demonstrates that our framework is able to learn multi-scale representations by our mutual learning scheme, which further boosts the performance.

**Compare with US-Net plus resolution.** As evident in Figure 1, applying different resolutions to US-Net during inference can already achieve improvements over the original US-Net. We denote this method as US-Net+. However, US-Net+ cannot achieve the optimal width-resolution balance due to lack of learning. In Figure 5, we plot the Accuracy-FLOPs curves of our framework and US-Net+ based on MobileNet v1 backbone, and highlight the selected input resolutions with different colors. We can see that as we decrease the FLOPs ($569 \rightarrow 468$ MFLPs), our framework first reduces network width to meet the constraint while keeping the $224\times224$ resolution (red color line in Figure 5). After 468 MFLOPs, our framework selects lower input resolution (192) and then continues reducing the width to meet the constraint. On the other hand, US-Net+ cannot find such balance. It always slims the network width and uses the same (224) resolution as the FLOPs decreasing until it goes to really low. This is because US-Net+ does not incorporate input resolution into the learning framework. Simply applying different resolutions during testing cannot achieve the optimal width-resolution balance.

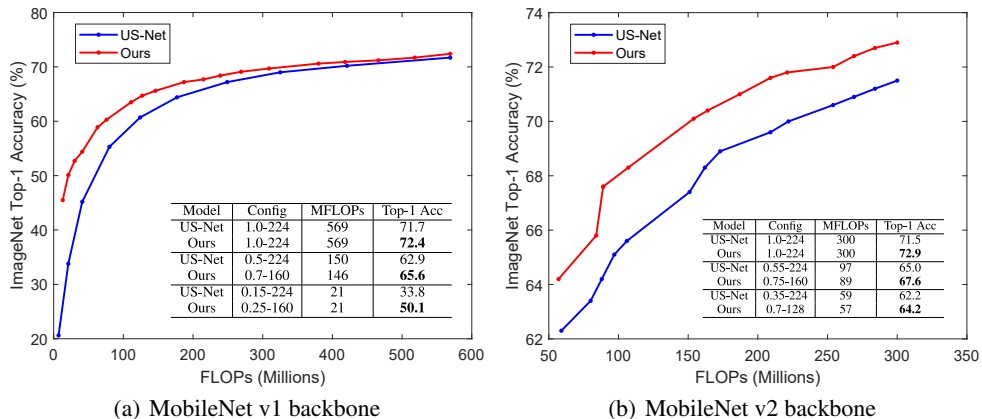

(a) MobileNet v1 backbone                    (b) MobileNet v2 backbone

Figure 4: Accuracy-FLOPs curves of our proposed framework and US-Net. (a) is based on MobileNet v1 backbone. (b) is based on MobileNet v2 backbone.

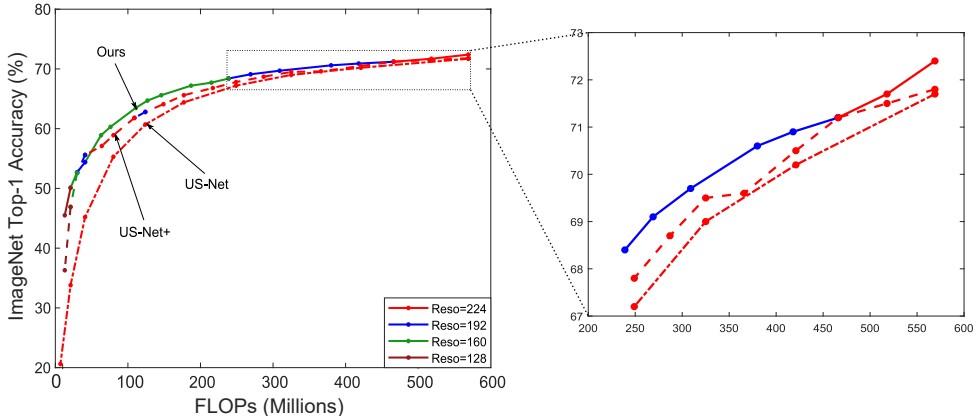

Figure 5: The Accuracy-FLOPs curve is based on MobileNet v1 backbone. We highlight the selected resolution under different FLOPs with different colors. For example, the solid green line indicates when the constraint range is [41, 215] MFLOPs, our framework constantly selects input resolution 160 but reduces the width to meet the resource constraint. Best viewed in color.

**Compare with independently trained networks.** To demonstrate that our framework is able to capture multi-scale representations for each sub-network, we compare with different width-resolution configurations. MobileNets (Howard et al., 2017; Sandler et al., 2018) have reported the results under different configurations. However, they consider width and resolution as independent factors, thus cannot take full advantage of the features contained in different resolutions. We compare the performance of our framework and independently-trained MobileNets in Figure 6. For MobileNet v1, widths are selected from $\{1.0\times, 0.75\times, 0.5\times, 0.25\times\}$, and resolutions are selected from $\{224, 192, 160, 128\}$, leading to 16 configurations in total. Similarly, MobileNet v2 selects configurations from $\{1.0\times, 0.75\times, 0.5\times, 0.35\times\}$ and $\{224, 192, 160, 128\}$. Our framework is executable at any FLOPs between the lower bound and upper bound. From Figure 6, our framework consistently outperforms MobileNets. Even at the same width-resolution configurations, we can achieve much better performance. This demonstrates that our framework not only finds the optimal width-resolution balance but also learns stronger representations by our mutual learning scheme.

Table 1: ImageNet Top-1 accuracy on MobileNet v1 backbone.

| Model | Best Configuration | FLOPs | Top-1 Acc |
|---|---|---|---|
| EfficientNet (Tan & Le, 2019) | $d = 1.4, w = 1.2, r = 1.3$ | 2.3B | 75.6% |
| Ours | $w = 1.6, r = 1.3$ | 2.3B | **77.1%** |

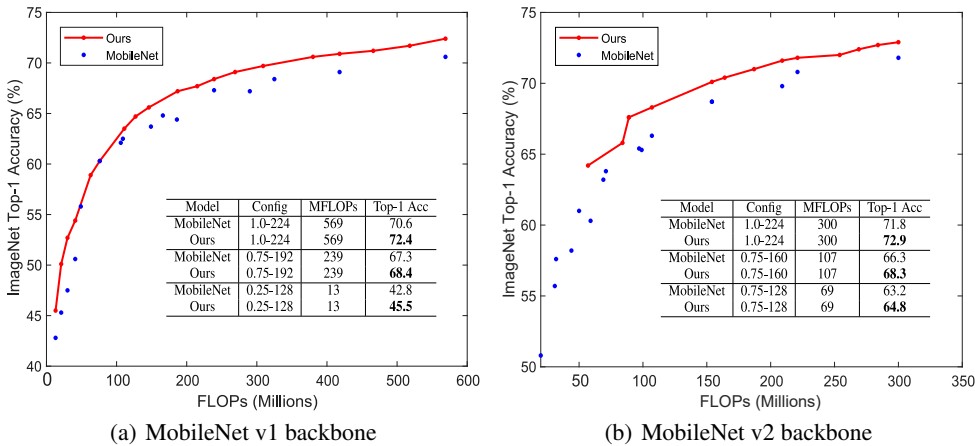

(a) MobileNet v1 backbone      (b) MobileNet v2 backbone

Figure 6: Accuracy-FLOPs curve of our framework and independently-trained MobileNets. (a) is MobileNet v1 backbone. (b) is MobileNet v2 backbone. The results for different MobileNets configuration are taken from the paper (Howard et al., 2017; Sandler et al., 2018)

To further validate the effectiveness of our mutual learning scheme, we compare our results with the state-of-the-art EfficientNet (Tan & Le, 2019). EfficientNet acknowledges the importance of balancing between network width, depth and resolution. But the authors also treat them as independent factors. They propose to grid search over these three dimensions to find the optimal configuration under certain constraint. Then the network can be scaled up to other constraints following this configuration. We compare with the best configuration they found for MobileNet v1 at 2.3 BFLOPs. To meet this constraint, we train our framework with a width range of $[1.0\times, 2.0\times]$, and select resolutions from $\{224, 256, 288, 320\}$. This makes our framework executable in the range of [0.57, 4.5] BFLOPs. The results are compared in Table 1. Our framework finds similar configuration as EfficientNet, except that we arrive at a larger width since depth is not considered. But our framework achieves significantly better performance than EfficientNet. We attribute this to the **mutual learning from width and resolution** scheme.

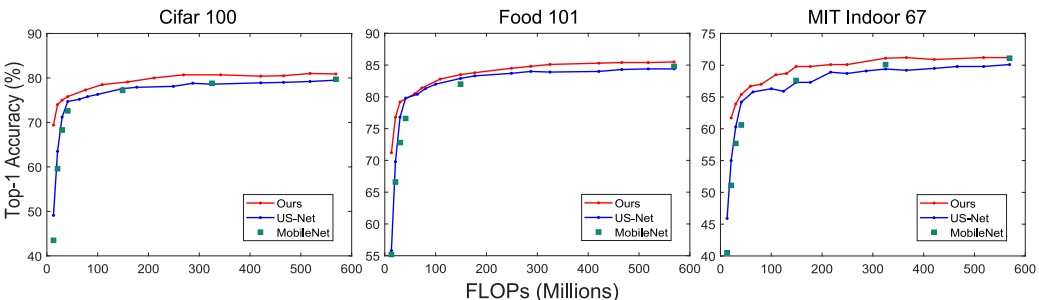

Figure 7: Accuracy-FLOPs curves on different transfer learning datasets.

## 4.2 TRANSFER LEARNING

To evaluate the representations learned by our framework, we further conduct experiments on three popular transfer learning datasets, Cifar-100 (Krizhevsky, 2009), Food-101 (Bossard et al., 2014) and MIT-Indoor67 (Quattoni & Torralba, 2009). Cifar-100 is superordinate-level object classification, Food-101 is fine-grained classification and MIT-Indoor67 is scene classification. Such large variety is suitable to evaluate the robustness of the learned representations. We compare our framework with US-Net and independently-trained MobileNet v1. We fine-tune all the models with a batch size of 256, initial learning rate of 0.1 with cosine decay schedule and a total of 100 epochs. Following the setting in ImageNet training, we use width range $[0.25\times, 1.0\times]$ and resolutions $\{224, 192, 160, 128\}$. The results are shown in Figure 7. Again, our framework achieves consistently

better performance than US-Net and MobileNet. This verifies that our framework is able to learn well-generalized representations.

### 4.3 OBJECT DETECTION AND INSTANCE SEGMENTATION

We also apply our framework to COCO object detection and instance segmentation (Lin et al., 2014). Our experiments are based on Mask-RCNN-FPN (He et al., 2017; Lin et al., 2017) and MMDetection (Chen et al., 2019a) toolbox. We use VGG-16 (Simonyan & Zisserman, 2014) as backbone network to validate our proposed mutual learning scheme.

We first pre-train VGG-16 on ImageNet using US-Net and our framework respectively. The network width range is $[0.25\times, 1.0\times]$ and resolutions are $\{224, 192, 160, 128\}$. Then we fine-tune the pre-trained models on COCO. The FPN neck and detection head are shared among different sub-networks. For simplicity, we don't use *inplace distillation*. Rather, each sub-network is trained with the ground truth. The other training procedures are the same as training ImageNet classification. Following common settings in object detection, US-Net is trained with image resolution $1000 \times 600$. Our framework randomly selects resolutions from $1000 \times \{600, 480, 360, 240\}$. All models are trained on COCO 2017 training set and tested on COCO 2017 validation set with different image resolutions. The Average Precision (AP) results for object detection and instance segmentation under different computational cost are presented in Figure 8. These results reveal that our framework significantly outperforms US-Net under all resource constraints. Specifically, on the full configuration (1.0×-600), US-Net achieves comparable results with independent VGG-16, while our framework performs much better. This again validates the effectiveness of our width-resolution mutual learning scheme. **Please refer to Appendix A.4 for visual comparison results.**

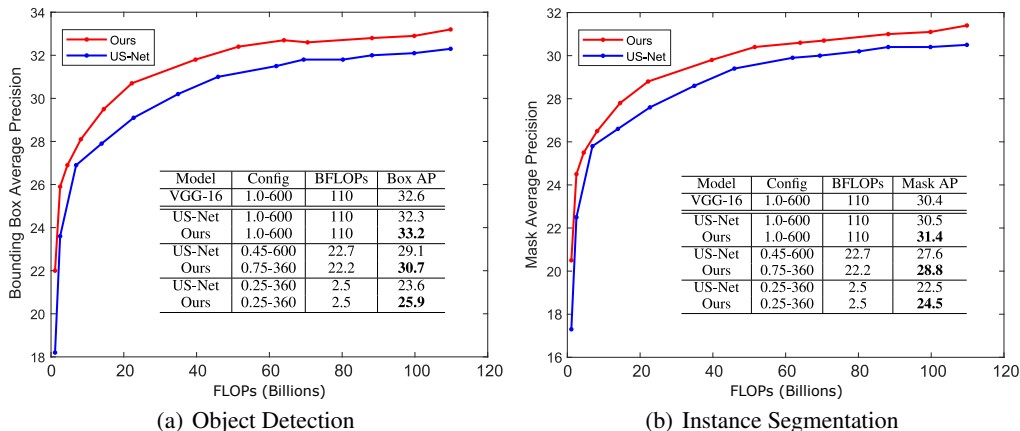

(a) Object Detection          (b) Instance Segmentation

Figure 8: Average Precision - FLOPs curves of our framework and US-Net. (a) is bounding box average precision. (b) is mask average precision.

## 5 CONCLUSION AND FUTURE WORK

This paper highlights the importance of simultaneously considering both network width and input resolution for designing efficient network structures. A new framework is defined capable of mutually learning from network width and input resolution. The proposed framework is demonstrated to significantly improve inference performance per FLOP for various datasets and tasks. The simplicity and generality of the framework allows it to translate well to generic problem domains. This also make logical extensions readily available by adding other network dimensions, e.g., network depth, to the framework. The framework can also be extended to video input and 3D neural networks, where we can leverage both spatial and temporal information.

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

# A   APPENDIX

## A.1   TRAINING DETAILS

**Hyperparameters.** We follow the training settings in US-Net (Yu & Huang, 2019) when training on ImageNet. The batch size is 1024 on 8 GPUs. Initial learning rate is 0.5 and decreases to 0 with a cosine decay schedule. The momentum is set to 0.9 and weight decay is 0.0001. All models are trained for 250 epochs using SGD optimizer.

**Network width scales.** In Figure 4, for MobileNet v1 backbone, our method uses width scale from $[0.25, 1]\times$ and resolutions from $\{224, 192, 160, 128\}$ while US-Net uses width scale from $[0.05, 1]\times$. For MobileNet v2 backbone, our method uses width scale from $[0.7, 1]\times$ and resolutions from $\{224, 192, 160, 128\}$ while US-Net uses width scale from $[0.35, 1]\times$. In transfer learning experiments, both our method and US-Net use width scale from $[0.25, 1]\times$. Specifically, for US-Net, we adopt the officially released model on width range $[0.25, 1]\times$ and finetune it with width range $[0.25, 1]\times$. In object detection and instance segmentation, we also utilize the same width range $[0.25, 1]\times$ for our method and US-Net.

The reason for using different width scales in Figure 4 is that both our approach and US-Net aim to train an adaptive network to meet the dynamic resource constraints at test time. We compared our approach and US-Net using the Accuracy-FLOPs curves under the same dynamic resource constraint. For MobileNet v1 backbone (Figure 4 (a)), the dynamic FLOPs constraint is [13, 569] MFLOPs. To meet this constraint, US-Net needs the width scale of $[0.05, 1]\times$. However, our approach can meet this computation constraint by balancing between width scale $[0.25, 1]\times$ and resolutions $\{224, 192, 160, 128\}$. Similarly, for MobileNet v2 (Figure 4 (b)), the dynamic constraint is [59, 300] MFLOPs. US-Net needs width scale of $[0.35, 1]\times$, while we can meet this constraint by combining width $[0.7, 1]\times$ and resolutions $\{224, 192, 160, 128\}$. Therefore, integrating resolution in our framework gives more flexibility to balance network width. This is one advantage that we illustrated in Figure 1.

In all the following experiments (US-Net+, transfer learning, object detection and instance segmentation), we use the same width scale $[0.25, 1]\times$ for both our method and US-Net. This setting is actually in favor of US-Net, because if the original US-Net$_{[0.05,1]}$ is used, which meets the same resource constraints as ours, it performs worse.

## A.2   CONTRIBUTION OF KL DIVERGENCE

In the experiments, we follow US-Net to train sub-networks using KL divergence to have a fair comparison. In US-Net, the author claims that training with KL divergence is better than training with ground truth label. In this section, we give some quantitative results to show the contribution of KL divergence to the overall performance. The training setting is the same as training on ImageNet, except that sub-networks are trained with the ground truth rather than the soft labels from the full-network. As shown in Figure 9, training with KL divergence (i.e., using soft labels) achieves marginal improvement over using the ground truth. The differences tend to be larger when FLOPs goes down. The reason might be that small sub-networks have limited learning capacity and it is easier to learn using soft labels from the teacher rather than using one-hot ground truth labels.

## A.3   VGG-16 IMAGENET PRE-TRAINING

We use VGG-16 as the backbone network for object detection and instance segmentation. We first pretrain VGG on ImageNet following US-Net and our framework. The batch size is set to 512. The initial learning rate is 0.01 and decreases to 0 with a cosine decay schedule. Both frameworks are trained for 100 epochs. The Accuracy-FLOPs curves are shown in Figure 10. We can see that our framework still constantly outperforms US-Net on such large neural network. This further demonstrates the effectiveness and generalization ability of our framework.

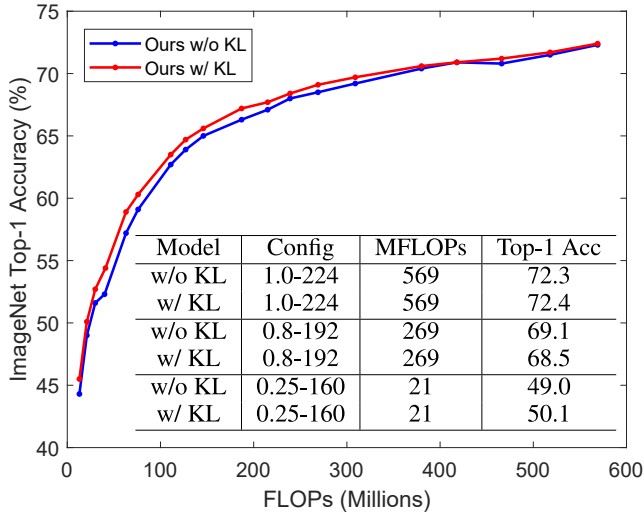

Figure 9: Accuracy-FLOPs curves of our framework w/ and w/o KL divergence using the MobileNet v1 backbone.

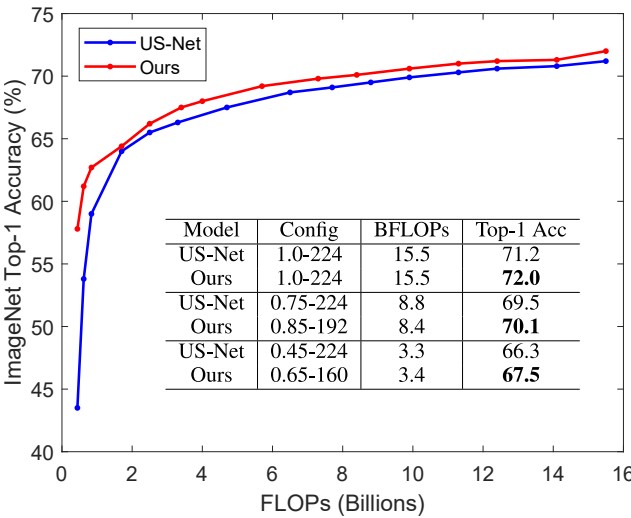

Figure 10: Accuracy-FLOPs curves of our framework and US-Net on VGG-16 backbone.

### A.4 Object detection and instance segmentation examples

We show some visualization examples in Figure 11. We tested two model configurations $1.0\times$-600 and $0.5\times$-480. The upper pairs are $1.0\times$-600, the lower pairs are $0.5\times$-480. From Figure 11, we can see that our framework is able to detect both small-scale and large-scale objects in both configurations while US-Net fails. This is because our mutual learning scheme enables each sub-network to capture multi-scale representations.

### A.5 Larger tables for better readability

In this subsection, we display a larger version of the tables in the paper for better readability. Table 2 and Table 3 show comparison results of our method and US-Net on ImageNet. Table 4 and Table 5 present results of our approach and independently-trained MobileNets on ImageNet. Table 6 and Table 7 compare the results on COCO object detection and instance segmentation.

Table 2: MobileNet v1 backbone

| Model | Config | MFLOPs | Top-1 Acc |
|-------|--------|--------|-----------|
| US-Net | 1.0-224 | 569 | 71.7 |
| Ours | 1.0-224 | 569 | **72.4** |
| US-Net | 0.5-224 | 150 | 62.9 |
| Ours | 0.7-160 | 146 | **65.6** |
| US-Net | 0.15-224 | 21 | 33.8 |
| Ours | 0.25-160 | 21 | **50.1** |

Table 3: MobileNet v2 backbone

| Model | Config | MFLOPs | Top-1 Acc |
|-------|--------|--------|-----------|
| US-Net | 1.0-224 | 300 | 71.5 |
| Ours | 1.0-224 | 300 | **72.9** |
| US-Net | 0.55-224 | 97 | 65.0 |
| Ours | 0.75-160 | 89 | **67.6** |
| US-Net | 0.35-224 | 59 | 62.2 |
| Ours | 0.7-128 | 57 | **64.2** |

Table 4: MobileNet v1 backbone

| Model | Config | MFLOPs | Top-1 Acc |
|-------|--------|--------|-----------|
| MobileNet | 1.0-224 | 569 | 70.6 |
| Ours | 1.0-224 | 569 | **72.4** |
| MobileNet | 0.75-192 | 239 | 67.3 |
| Ours | 0.75-192 | 239 | **68.4** |
| MobileNet | 0.25-128 | 13 | 42.8 |
| Ours | 0.25-128 | 13 | **45.5** |

Table 5: MobileNet v2 backbone

| Model | Config | MFLOPs | Top-1 Acc |
|-------|--------|--------|-----------|
| MobileNet | 1.0-224 | 300 | 71.8 |
| Ours | 1.0-224 | 300 | **72.9** |
| MobileNet | 0.75-160 | 107 | 66.3 |
| Ours | 0.75-160 | 107 | **68.3** |
| MobileNet | 0.75-128 | 69 | 63.2 |
| Ours | 0.75-128 | 69 | **64.8** |

Table 6: Object Detection

| Model | Config | BFLOPs | Box AP |
|-------|--------|--------|--------|
| VGG-16 | 1.0-600 | 110 | 32.6 |
| US-Net | 1.0-600 | 110 | 32.3 |
| Ours | 1.0-600 | 110 | **33.2** |
| US-Net | 0.45-600 | 22.7 | 29.1 |
| Ours | 0.75-360 | 22.2 | **30.7** |
| US-Net | 0.25-360 | 2.5 | 23.6 |
| Ours | 0.25-360 | 2.5 | **25.9** |

Table 7: Instance Segmentation

| Model | Config | BFLOPs | Mask AP |
|-------|--------|--------|---------|
| VGG-16 | 1.0-600 | 110 | 30.4 |
| US-Net | 1.0-600 | 110 | 30.5 |
| Ours | 1.0-600 | 110 | **31.4** |
| US-Net | 0.45-600 | 22.7 | 27.6 |
| Ours | 0.75-360 | 22.2 | **28.8** |
| US-Net | 0.25-360 | 2.5 | 22.5 |
| Ours | 0.25-360 | 2.5 | **24.5** |

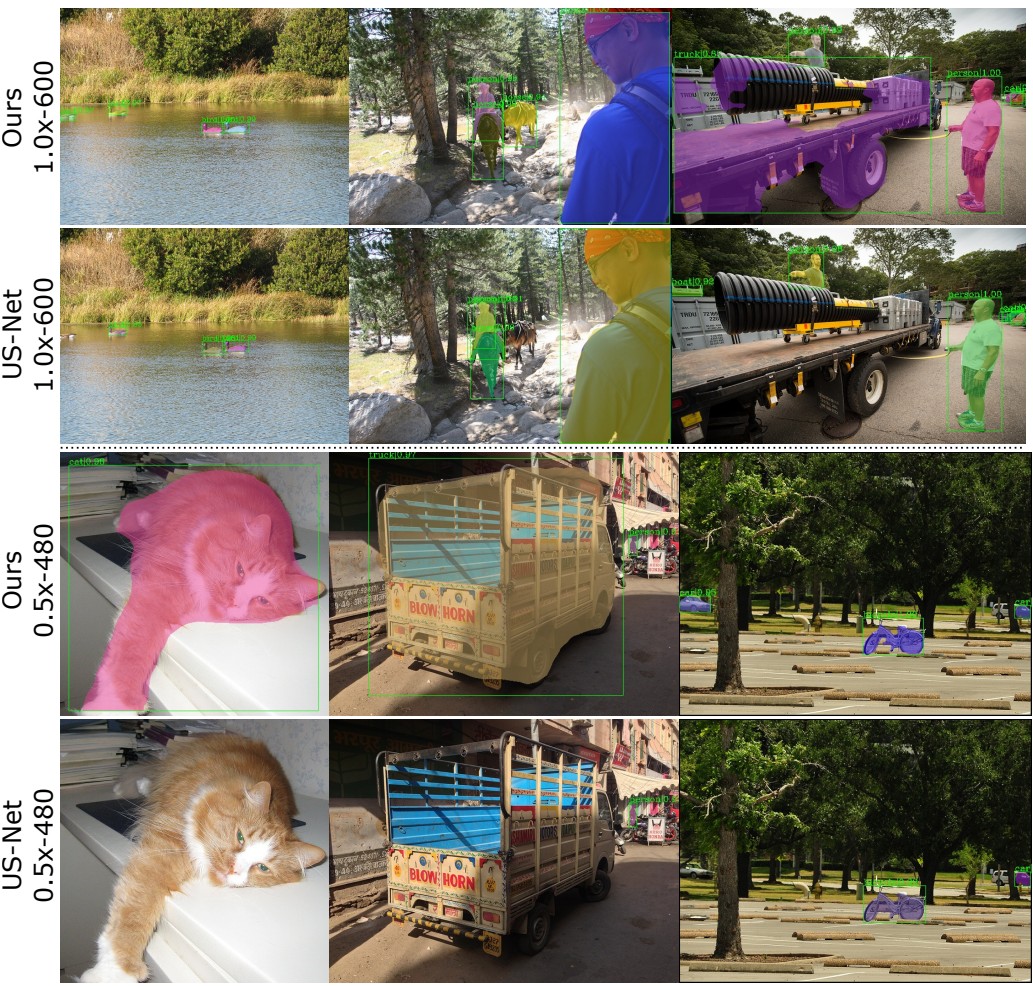

Figure 11: Object detection and instance segmentation examples.

