# OpenReview forum: "A closer look at network resolution for efficient network design"
_ICLR.cc/2020/Conference — Reject_

### Official Review · AnonReviewer2 · 2019-10-23
**Official Blind Review #2**

**Rating:** 6

**Review:**

The paper explore how varying input image resolution can have a powerful effect in reducing computational complexity of Convolutional NN's to the point of outperforming state of the art efficient architectures.
More in detail, the paper proposes a method of joint training of multiple resolutions networks, leveraging student/teacher/distillation from scratch. This is based on training a high resolution teacher network and a low resolution student network, as well as a number of intermediate resolutions networks sampled randomly and jointly during training. Thanks to distillation well known regularization effects, the proposed method is achieving competitive results compared to existing state of the art efficient network architectures. The authors claim, and to some extent show, that this is due to the ability of the proposed method to take into account in a optimal way multi-resolution features available in the image. The paper is well written and presented with extensive results, comparing computational complexity/accuracy curves to existing state of the art architectures, as well as results on transfer learning to show that the feature learned do indeed generalize and don't necessarily overfit to imagenet. The idea is rather simple, but the results and the execution is inspiring.

**Experience Assessment:**

I have published in this field for several years.

**Review Assessment: Checking Correctness Of Derivations And Theory:**

N/A

**Review Assessment: Checking Correctness Of Experiments:**

I assessed the sensibility of the experiments.

**Review Assessment: Thoroughness In Paper Reading:**

I read the paper thoroughly.

---

> ### Author Response · Authors · 2019-11-08
> **Response to reviewer 2**
>
> We would like to sincerely thank you for the comments and appreciation of our work. Our work is the first to consider both network structure (i.e., network width) and input resolution in a unified learning framework to achieve an adaptive network that can instantly tradeoff between accuracy and latency at run time. Since the input scale is not well explored in previous work for achieving better accuracy-efficiency tradeoffs, our work strives to close the gap between network structure and network input. The mutual learning from different network widths and input resolutions is novel and our extensive experiments demonstrate its effectiveness over different network structures, datasets as well as tasks. The generality of this framework also makes it logically ready to extend to video input and 3D neural networks, where we can leverage both spatial resolution and temporal resolution. We believe our work brings new insights for resource-adaptive network/framework design.

---

### Official Review · AnonReviewer1 · 2019-10-24
**Official Blind Review #1**

**Rating:** 3

**Review:**

This paper proposes a multi-resolution training scheme for a slimmable network. The proposed method provides a new regime leveraging diverse image resolutions for training sub-networks, which enables efficient network scaling. Throughout the experiments, the authors claim the proposed method shows better performance compared to US-net.

Pros)
(+) The idea of multi-resolution training combining to a slimmable network looks good
(+) Applying a slimmable network's technique to other tasks including detection and segmentation looks good
(+) The combination of multi-resolution and the slimmable network seems to be reasonable.
(+) The paper is well written and looks justified well.
(+) The authors provided extensive experiments.

Cons)
(-) There is no backups why the proposed method could outperform over US-net.
(-) The proposed method is incremental and improvements are marginal.
(-) Looks like there exists missing in details of the experiments.
(-) The performance report of the compared methods is quite strange.

Comments)
- The proposed method is too straightforward, so the authors should clarify why it works over US-net. Additionally, can the authors provide advantages using a different image-scale need for training a different sub-network?
- The authors should clarify the training details of US-Net used in this paper. The performance of US-Net in Figure 4 (a) looks the same as the performance of US-Net trained with [0.05, 1]x scaling in the original paper. However, in the original paper, the authors of US-net reported [0.05, 1]x scaling as the worst performance setting in the original paper. Therefore, the authors should compare their method with the best performance setting of US-Net, which is [0.25, 1]x (because the proposed method looks being used [0.25, 1]x training setting, so the comparison should be done in fair).
- The scaling parameters of US-Net used in the experiments should be specified. All the results of US-Net do not contain where they come from (i.e., the training width bound in US-net).
- Can the authors report the results for 0.5-224 and 0.15-224 in Figure.4(a)? Why 0.7-160 and 0.25-160 were picked?
- In Table 1, the performance of EfficientNet is weird. EfficientNet-B2 has 79.8% accuracy with 1.0B FLOPs, but the reported performance in this paper of EfficientNet has 75.6% accuracy with 2.3B FLOPs. Please clarify this.
- How much does KLdiv contribute to the overall performance?
- All the tables are not clearly shown. Please reattach all the tables for better readability.

About rating)
I think the idea looks novel, but the method is quite straightforward, and the paper does not incorporate any analysis as a backup for the proposed method. The initial rating is towards reject, but I would like to see the authors' response and the other reviewers' comments. After that, the final rating might be changed.

**Experience Assessment:**

I have published one or two papers in this area.

**Review Assessment: Checking Correctness Of Derivations And Theory:**

I assessed the sensibility of the derivations and theory.

**Review Assessment: Checking Correctness Of Experiments:**

I carefully checked the experiments.

**Review Assessment: Thoroughness In Paper Reading:**

I read the paper at least twice and used my best judgement in assessing the paper.

---

> ### Author Response · Authors · 2019-11-08
> **Response to reviewer 1 (Part 1)**
>
> We would like to sincerely thank you for the detailed comments. We addressed each question as follows. We reorganize the order of these questions for better explanations.
>
> Q1. The scaling parameters of US-Net used in the experiments should be specified.
>
> A: We apologize for missing some training details in the paper. In the experiments of ImageNet classification, for MobileNet v1 backbone, our method uses width scale from [0.25, 1] and resolutions from {224, 192, 160, 128} while US-Net uses width scale from [0.05, 1]. For MobileNet v2 backbone, our method uses width scale from [0.7, 1] and resolutions from {224, 192, 160, 128} while US-Net uses width scale from [0.35, 1]. In transfer learning experiments, both our method and US-Net use width scale from [0.25, 1]. Specifically, for US-Net, we adopt the officially released model on width range [0.25, 1] and finetune it with width range [0.25, 1]. In object detection and segmentation, we also use the same width range [0.25 ,1] for our method and US-Net. We add these details in Appendix 1 in the updated version.
>
> Q2. The authors should compare their method with the best performance setting of US-Net, which is [0.25, 1]x.
>
> A: We would like to clarify that our experimental settings are fair. Given that both our approach and US-Net aim to train an adaptive network to meet the dynamic resource constraints at test time. We compared our approach and US-Net using the Accuracy-FLOPs curves under the same dynamic resource constraint. For MobileNet v1 backbone (Figure 4 (a)), the dynamic FLOPs constraint is [13, 569] MFLOPs. To meet this constraint, US-Net needs the width scale of [0.05, 1]x. However, our approach can meet this computation constraint by balancing between width scale [0.25, 1]x and resolutions {224, 192, 160, 128}. In other words, by considering input resolution, we can set the training network width scale to [0.25, 1]x (no need to go low as 0.05x of width) in order to meet the resource constraint (i.e., [13, 569] MFLOPs). Similarly, for MobileNet v2 (Figure 4 (b)), the dynamic constraint is [59, 300] MFLOPs. US-Net needs width scale of [0.35, 1]x, while we can meet this constraint by combining width [0.7, 1]x and resolutions {224, 192, 160, 128}. Therefore, integrating resolution in our framework gives more flexibility to balance network width. This is one advantage that we illustrated in Figure 1.
> In our experiments, we also compared to US-MobileNet v1 trained with width [0.25, 1]x. We even made one step further, that is to compare to our proposed US-Net+ as illustrated in Figure 5. US-Net+ is applying different resolutions {224, 192, 160, 128} to US-Net$_{[0.25, 1]}$ during test time. We choose the best-performing width-resolution configurations under different FLOPs to get its Accuracy-FLOPs curve, which is the upper bound of US-Net$_{[0.25, 1]}$. As demonstrated in Figure 5, our method outperforms US-Net+ because US-Net+ can not find the optimal width-resolution balance due to lack of multi-resolution learning. Our method also outperforms US-Net$_{[0.25, 1]}$ and achieves wider dynamic constraints. Besides, as clarified in Q1, for all the experiments in transfer learning, object detection and segmentation, both our method and US-Net use the same width scale [0.25, 1]. This setting is actually in favor of US-Net, because if the original US-Net$_{[0.05, 1]}$ is used, which meets the same resource constraints as ours, it performs worse.
>
> Q3. The proposed method is too straightforward, so the authors should clarify why it works over US-net. Additionally, can the authors provide advantages using a different image-scale need for training a different sub-network?
>
> A: (1) --about the method
> We would like to clarify that the proposed method is not straightforward. As discussed in Section 2, most previous works focus on reducing computational cost only from the structure level, ignoring the importance of input scale for achieving better accuracy-efficiency tradeoffs. Recently, EfficientNet [1] points out that balancing between network depth, width and resolution can lead to better performance, but it only considers network depth, width and resolution as independent factors by simply searching over different depth-width-resolution configurations. In our work, we try to bridge the gap between network structure and network input. In the paper, we first shed light on the promising advantages of taking input resolution into account for achieving better accuracy-efficiency in Section 2. In light of this, we proposed a unified framework to learn from different network widths and input resolutions jointly, i.e., deep mutual learning in our framework. As illustrated in Figure 5, trivially applying different resolutions during test time can not achieve the optimal tradeoffs, while our joint training framework consistently improves the performance on different backbones, datasets and tasks.
>
> (continued in part 2)

---

> > ### Author Response · Authors · 2019-11-08
> > **Response to reviewer 1 (Part 2)**
> >
> > (2) --advantages of using different image scale need for training a different sub-network?
> >
> > Each sub-network in our framework randomly choose an image resolution from {224, 192, 160, 128}. As explained in Section 2, adjusting input resolution is an effective way to achieve accuracy-efficiency tradeoffs. Our method is able to find better width-resolution balance under different resource constraints since different width-resolution configurations are visited in the training framework. It should be noted that this is not naturally achievable by simply applying different input resolutions during testing as illustrated in Figure 5.
> > Moreover, different resolutions contain different information [2, 3]. Lower resolution images may contain more global structures while higher resolution ones may encapsulate more fine-grained patterns. Our method learns multi-scale representations from different resolutions during training. We explained this in Section 3.2 and validated it by comparing to MobileNet v1, v2 under the same width-resolution configuration. Our method consistently outperforms MobileNet v1 and v2 at different width-resolution configurations, even if it may not be the best-performing configuration in our framework. We further compared to the optimal configuration searched by EfficientNet to show that our method learns better representations. In Appendix A.3, we also give some qualitative object detection and instance segmentation results to show that our method is more robust to both small-scale and large-scale objects, thanks to the multi-scale representation learned from different input resolutions.
> >
> > Q4. The improvements are marginal.
> >
> > A: Note that we aim to train adaptive networks to meet dynamic constraints, so we need to look at the whole Accuracy-FLOPs curve rather than picking some particular points on the curve. In Figure 4, our framework outperforms US-Net by 1%-2% over a large spectrum of FLOPs. As the FLOPs goes down, our method achieves significant improvements (e.g., 33.8% vs 50.1% at 21 MFLOPs). The performance in such low computational resource is usually more meaningful/important in real-world applications, such as mobile devices are at low battery or they are running multiple Apps at the same time. In object detection and segmentation tasks, our method also consistently outperforms US-Net by 1-2 mAP, which is considered as a significant improvement in these challenging tasks.
> > Another point to note is that we proposed a general training framework. The generality and effectiveness of our method makes it applicable for various networks structures, datasets and tasks. It is a plug-and-play strategy and achieves competitive improvements without bells and whistles. The simplicity also makes it logically ready to extend to video input and 3D neural networks, where we can leverage both spatial resolution and temporal resolution.
> >
> > Q5. Can the authors report the results for 0.5-224 and 0.15-224 in Figure.4(a)? Why 0.7-160 and 0.25-160 were picked?
> >
> > A: By considering both network width and input resolution in our learning framework, our method can obtain different width-resolution configurations for the same FLOPs. We choose the best-performing configuration because all these configurations are executable during test time. For ~150 MFLOPs it is 0.7-160, for ~21 MFLOPs it is 0.25-160. The result for 0.5-224 is 63.1% with 150 MFLOPs. But we can achieve 65.6% with 146 MFLOPs using the 0.7-160 configuration. It does not make sense to do 0.15-224 because our width scale is [0.25, 1], which is explained in the response to Q2.
> >
> > Q6. In Table 1, the performance of EfficientNet is weird.
> >
> > A: EfficienNet-B2 is based on a different backbone searched by NAS techniques. We are comparing to the same MobileNet v1 backbone. The result is taken from Table 3 in the original paper [1].
> >
> > Q7. How much does KLdiv contribute to the overall performance?
> >
> > A: We follow the KLdiv setting in the US-Net to have a fair comparison. In US-Net, the authors claim that the KLdiv can give better performance than only training with the ground truth, but unfortunately the authors didn’t provide any quantitative results. We are glad to give some quantitative results and analyses on this. We will update our results after finishing the experiments.
> >
> > Q8. All the tables are not clearly shown. Please reattach all the tables for better readability.
> >
> > A: We have reattached a larger version of these tables in the Appendix. Please check.
> >
> > References:
> > [1] Tan, Mingxing, and Quoc Le. "EfficientNet: Rethinking Model Scaling for Convolutional Neural Networks." International Conference on Machine Learning. 2019.
> > [2] Chen, Yunpeng, et al. "Drop an octave: Reducing spatial redundancy in convolutional neural networks with octave convolution." arXiv preprint arXiv:1904.05049 (2019).
> > [3] Chin, Ting-Wu, Ruizhou Ding, and Diana Marculescu. "Adascale: Towards real-time video object detection using adaptive scaling." arXiv preprint arXiv:1902.02910 (2019).

---

> > > ### Author Response · Authors · 2019-11-13
> > > **Updated results and more clarifications**
> > >
> > > Q7. How much does KLdiv contribute to the overall performance?
> > >
> > > A: We updated the experimental results in Appendix 2 of the revised paper. The result shows that KLdiv only achieves marginal improvements to the overall performance, which is consistent with the claims in US-Net. The differences tend to be larger when FLOPs goes down. The reason might be that small sub-networks have limited learning capacity and it is easier to learn using soft labels from the teacher rather than using one-hot ground truth labels.
> > >
> > > Q. why the proposed method could outperform over US-net?
> > >
> > > A: As stated at the end of Section 2 in the paper, our framework could outperform US-Net because (a) we can achieve better width-resolution balance as illustrated in Figure 5 (best viewed in color), (b) we can capture multi-scale representations as demonstrated in comparison to independently-trained MobileNets and EfficientNet, and visual examples in detection and segmentation. Detailed explanations are in the response to Q.3-(2).

---

### Official Review · AnonReviewer4 · 2019-11-13
**Official Blind Review #4**

**Rating:** 3

**Review:**

The authors propose a new training regime for multi-resolution slimmable networks. Their approach is based on US-Net training technique but in addition to sampling from different network widths they also sample from different input resolutions and show that using multi-scale inputs improves the top-1 accuracy on ImageNet comparing to US-Net or MobileNet v1/v2 within the same resource constraints.

Pros:
+ The authors correctly identify input resolution as one of the aspects of lightweight network design that is often overlooked
+ They propose a practically viable training scheme that can be used to train & select networks given resource constraints
+ The paper is well written and includes many insightful experimental findings

Con:

The authors specify the mutual learning from width and resolution as their main contribution. They insist that treating input resolution independently from network structure is what distinguishes previous work from the newly suggested technique. But the paper doesn't include extensive experimental comparisons with the approaches that treat input resolution independently. Thus its claim that joint width/resolution sampling is beneficial comparing to independent approaches is somewhat unfounded.

For example, the authors show that MobileNet with 1.0-224 config (no sampling from widths nor from input resolutions during training) is outperformed by their network with 1.0-224 config (which effectively samples only from input resolutions during training). This is not surprising as one can view sampling from input resolutions as an equivalent to data augmentation. The importance of data augmentation is well known, so to prove the proposed mutual learning is beneficial the authors would need to compare against the networks that were trained using this multi-scale data augmentation. Figure 5 has a similar comparison but the only multi-resolution baseline there is US-Net+ which isn't using multi-resolution images in training. The paper would greatly benefit from adding such comparisons and proving they are not marginal.

On rating:

I'd summarize the idea of this paper as A) US-Net + B) multi-scale data augmentation + C) selecting the best network based on both input resolution and width to achieve optimal performance within resource constraints. Although C is practical and novel contribution, it is also quite straightforward. I would like to see authors response on how their approach differs from US-Net + multi-scale data augmentation for training and how/why this works better.

**Experience Assessment:**

I have read many papers in this area.

**Review Assessment: Checking Correctness Of Derivations And Theory:**

I assessed the sensibility of the derivations and theory.

**Review Assessment: Checking Correctness Of Experiments:**

I carefully checked the experiments.

**Review Assessment: Thoroughness In Paper Reading:**

I read the paper thoroughly.

---

> ### Author Response · Authors · 2019-11-14
> **Response to Reviewer 4**
>
> We would like to sincerely thank the reviewer for the comments. In our experiments, both US-Net and MobileNets are trained with the multi-scale data augmentation indicated by the reviewer. It is implemented by ‘transforms.RandomResizedCrop(224, scale=(0.25, 1.0))’ in the codes. This means that an image of random ratio/scale (0.25 to 1.0) of the original size is cropped and then resized to a random aspect ratio (3/4 to 4/3) of the original aspect ratio. The crop is finally resized to the given size (224). Therefore, this is a multi-scale data augmentation process and our method outperforms US-Net + multi-scale augmentation.
>
> On the other hand, our multi-resolution TRAINING is not equivalent to multi-scale data augmentation (which can be considered as a pre-processing step). Our framework randomly feeds different resolution images to different sub-networks. It has the following advantages over multi-scale data augmentation. (a) Smaller resolutions can reduce computational cost. Therefore, we don’t have to go to an extremely small network width to meet the resource constraints (please see the detailed explanation in the response to Reviewer 1, Q2). (b) Multi-resolution training helps find better width-resolution balance while US-Net trained with multi-scale data augmentation fails because there is no width-resolution learning in the multi-scale data augmentation scheme. (c) Since our sub-networks share weights with each other, different sub-networks can share the representations learned from different resolutions, enabling each sub-network to capture multi-scale representations as illustrated in Figure 3. The multi-scale representation learning has been proven effective in previous works [1, 2, 3] and it is not contradictory or equivalent to multi-scale data augmentation. Specifically, as pointed out by the reviewer, our framework outperforms MobileNets at 1.0-224 config. As illustrated in Figure 2, the 1.0-224 config is the full-network and is trained with the 224 resolution only, so the improvement is coming from the shared multi-scale representations with other sub-networks as discussed above.
>
> [1] Chen, Yunpeng, et al. "Drop an octave: Reducing spatial redundancy in convolutional neural networks with octave convolution." arXiv preprint arXiv:1904.05049 (2019).
> [2] Sun, Ke, et al. "Deep High-Resolution Representation Learning for Human Pose Estimation." Proceedings of the IEEE Conference on Computer Vision and Pattern Recognition. 2019.
> [3] Lin, Tsung-Yi, et al. "Feature pyramid networks for object detection." Proceedings of the IEEE conference on computer vision and pattern recognition. 2017.

---

> > ### Comment · AnonReviewer4 · 2019-11-14
> > **Please read "multi-scale data augmentation" as "multi-resolution data augmentation"**
> >
> > Thanks for the clarification! To avoid confusion let's stick to authors definitions of multi-resolution training and multi-scale data augmentation. Multi-scale data augmentation (where you scale back to full resolution) is definitely not equivalent to resolution sampling proposed in the paper and this is not what I had in mind. In paper and in comments "multi-scale" and "multi-resolution" were used interchangeably but it's good to keep them separate.
> >
> > Whenever you read "multi-scale data augmentation" in the review above, please read it as "multi-resolution data augmentation". I argue that multi-resolution training is not different from multi-resolution data augmentation. Multi-resolution data augmentation can also be applied as a preprocessing step and (depending on network design of course) may not require much changes to the training process except-  perhaps - for grouping inputs into batches by resolution. With such multi-resolution data augmentation accuracy for proposed 1.0 configs should be similar to 1.0 configs of independently trained networks as (supposedly) they should be fed with the same input data and should use the same network width. Inplace distillation also shouldn't interfere as it doesn't apply to 1.0 configs.
> >
> >
> > The paper would benefit from comparing:
> >
> > A) Independently trained networks (1.0-x configs for MobileNet, etc) - those are present in the paper already
> > B) Those same networks trained with multi-resolution inputs - I argue that for full width models those should be close to 1.0-224 authors configs from Figure 6
> > C) US-Net trained with multi-resolution inputs
> > D) Authors proposal
> >
> > The difference between C) and D) would be that in one step in D) sub-networks may potentially see different resolutions while in C) they all should see the same resolution. Showing that D > C will indicate that mutual learning from width and resolution is beneficial while seeing that C is comparable to D will indicate that one can consider widths and resolutions separately.

---

> > > ### Author Response · Authors · 2019-11-15
> > > **Response to 'multi-resolution data augmentation' (Part 1)**
> > >
> > > Thanks for the clarification.
> > >
> > > 1. US-Net trained with multi-resolution inputs --- method C)
> > >
> > > We would like to argue that the described US-Net + multi-resolution training is a special case of our training scheme. Specifically, the described US-Net + multi-resolution training can take on two settings. (a) One setting is that the full width network is trained with the 224 resolution and the other sub-networks adopt the same but a randomly sampled resolution in each iteration. This is just a special case of our method and the widths and resolutions are not treated independently in this case. This is because: first, different width-resolution configurations are still trained in the framework and contribute to updating the weights of different sub-networks. Second, sub-networks still see different resolutions if they sample a different resolution from that of the full network. (b) The other setting is that both the full-network and other sub-networks adopt the same but randomly sampled resolution. Again, this is another special case of our training framework as discussed above. Moreover, As stated at the end of the ‘training framework’ section of the paper, we conducted experiments where the full network also randomly sample different resolutions during training, but it yields worse performance. We argue that the full network has the largest learning capacity and is supposed to capture more detailed discriminatory features from the highest resolution. This also indicates our multi-width and resolution learning framework is not straightforward and has specific design choice.
> > > In summary, the described “US-Net + multi-resolution training” is a special case of our training scheme (as explained in (a) and (b)), and it does not consider widths and resolutions separately. Therefore, comparing “C) US-Net trained with multi-resolution inputs” and “D) Authors proposal” can not reach any conclusion stated by the reviewer, i.e. “Showing that D > C will indicate that mutual learning from width and resolution is beneficial while seeing that C is comparable to D will indicate that one can consider widths and resolutions separately.”
> > >
> > > 2. MobileNets (1.0-x)  trained with multi-resolution  --- method B)
> > >
> > > (1). The reviewer claims that ‘multi-resolution data augmentation’ could improve the performance of independent network, however, to the best of our knowledge, there is no previous literatures that adopt such data augmentation and demonstrate it is effective in classification. The reviewer claims the performance of Method B should be similar to our 1.0-224 config because ‘they should be fed with the same input data and should use the same network width’. This is not true. ‘Independent network + multi-resolution’ only has a fixed width, the weights are optimized towards the same resolution direction in each iteration. However, in our framework, we randomly sample another three sub-networks which share the weights with 1.0x. Since different sub-networks are optimized towards different resolution directions, the different parts of the weights of 1.0x are optimized towards a different (mixed) resolution direction as illustrated in Figure 3. Therefore, multi-resolution data augmentation is not equivalent to our width-resolution mutual learning framework.

---

> > > > ### Author Response · Authors · 2019-11-15
> > > > **Response to 'multi-resolution data augmentation' (Part 2)**
> > > >
> > > >
> > > > (2). The objective and motivation of our framework and multi-resolution augmentation is different. The “multi-resolution data augmentation” aims to improve the accuracy of a specific network. However, the multi-resolution learning in our framework aims to train an adaptive network that can execute in a spectrum of computing constraints/budgets (measured by FLOPs) at runtime. In this context, the multi-resolution learning aims to balance between width and resolution for different computing constraints during testing. We could use the “multi-resolution data augmentation” strategy to train a network. But we have to train multiple networks (e.g., Mobilenet-1.0x, Mobilenet-0.9x, Mobilenet-0.8x, …) in order to meet different computing constraints. This is not a scalable solution and requires storing all these models for deployment, while our solution only involves a single adaptive network by our multi-width and resolution training scheme.
> > > > Moreover, because of the objective of adapting to various computing budgets, our framework has a range for the network width, for example [0.25x, 1.0x] and 0.25x is the lower bound of the network width.  A smaller lower bound will yield worse performance for the same config while a higher lower bound will achieve better performance. For example, on MobileNet v2, we conducted two experiments with different width ranges ([0.35, 1.0] and [0.7, 1.0]) but the same resolutions {224, 192, 160, 128}. The performance of the 1.0-224 config using width range [0.35, 1.0] and [0.7, 1.0] is 72.0 vs 72.9. Therefore, we can not have a fair comparison between Method-B and our proposal. Our approach considers a dynamic resource constraint and the performance varies with the constraints, while MobileNet 1.0x trained with multi-resolution only aims at boosting the performance for the specific width.
> > > > But we still think this is a constructive advice and we are conducting the ‘Independent network + multi-resolution data augmentation’ experiment. Due to limited time, we are not able to finish the experiment before the discussion deadline. We report what we have by now below
> > > >
> > > > Epoch             |  1   |   5   | 10  | 15  |  20 |  25 |  30 |
> > > > Method-B      |21.2|42.1|48.5|50.5|51.1|51.9|52.3|
> > > > Ours-[0.7,1]   |23.1|44.7|50.1|53.3|53.1|54.6|54.7|
> > > >
> > > > The training settings are the same as ours, except that the width is fixed (1.0x) and only randomly sample resolution from {224, 192, 160, 128}. We know the above result does not represent the final performance, but it gives the trend of Method-B and our framework. The result demonstrates that our framework has a clear advantage over multi-resolution data augmentation.

---

### Public Comment · ~Hunting_Hunting1 · 2019-10-26
**Simple technique and no good explanation**

Hi there,
     I have read the paper and find that this paper proposes a simple technique to train different size cnns. I have several question.
     1. The paper claims that different input size of pictures can provide different knowledge, how can you get that conclusion?  I am not convinced by it. Just by the experimental results and draw the conclusion in a reverse way?
     2. What is the difference using the above techniques with knowledge distillation model. I didn't see much difference actually. Can you comment on it? Since from my side, optimize the KL loss between the big and smaller neural network is like knowledge distillation.
     3. The goal is to reduce FLOPs, and why there is no comparison with filter pruning method? There a bunch of work working on filtering pruning method that can achieve very good performance? So what is the advantage of this approach over previous ones. It lacks of comparison.

---

> ### Author Response · Authors · 2019-10-27
> **Response**
>
> Hi, thanks for your interest.
> 1. As discussed in the Section 2, multi-scale representations have been proven effective in various tasks and applications [1, 2, 3]. In [3], the authors reveal that higher resolution images contain more high frequency information which are usually encoded with fine details, while lower resolution images are usually encoded with global structure. Also, [4] points out that down-sampling the image can reduce false positives by avoiding focusing on unnecessary details in object detection.
> Nevertheless, previous works usually learn multi-scale representations by a multi-branch structure where each branch has a different spatial resolution. But the multi-branch structure needs careful design and is unfriendly to practical parallelization [5]. Our proposed framework is able to learn multi-scale representations without introducing such multi-branch structure thanks to our ‘mutual learning from width and resolution’ strategy.
> 2. The ‘inplace distillation’ is not the main point of our method. It is a training technique proposed in US-Net [6] and we follow it to have a fair comparison. The difference between ‘inplace distillation’ and vanilla knowledge distillation is that the teacher comes for free in ‘inplace distillation’. That is the bigger network can be used to teach smaller networks without introducing an additional network. In our framework, we train different sub-networks with different input scales. By sharing weights, all the sub-networks can share their knowledge with each other, enabling each sub-network to learn multi-scale representations. Furthermore, since we consider both width and resolution in the training process, our framework is able to find better width-resolution balance under different resource constraints during test time. This is not naturally achievable by simply applying different input resolutions during testing as illustrated in Figure 5.
> 3. The goal of our work is not to reduce FLOPs. Considering the varying resource constraints in real-life applications, we aim to develop a network that can be adaptively deployed under arbitrary resource constraints on the fly. Here, we use the number of FLOPs as the metric for resource constraint, which is widely used. To achieve this goal, we explore the importance of input scale for achieving better Accuracy-Efficiency tradeoffs at runtime. As discussed in Section 2, the input resolution is rarely considered in designing efficient structures, but it has several advantages in achieving better Accuracy-Efficiency tradeoffs. Accordingly, we proposed a training framework which is able to find better width-resolution balance under different resource constraints and learn multi-scale representations from different input scales. It should be noted that our work is orthogonal to pruning methods. It can be applied to pruned models/networks. More importantly, our framework is executable at arbitrary resource constraints during test time, without the need of re-training. However, re-training is usually necessary for pruning based approaches. We explained this clearly in Section 2. In the experiments, we compare to US-Net because we have the same goal. We compare to independently-trained networks under the same width-resolution configurations to demonstrate the effectiveness of our learned multi-scale representations. We further conduct experiments in popular transfer learning datasets and object detection and instance segmentation to demonstrate the generalization ability of our framework.
>
> We hope these can answer your questions.
>
> References:
> [1]Lin, Tsung-Yi, et al. "Feature pyramid networks for object detection." Proceedings of the IEEE conference on computer vision and pattern recognition. 2017.
> [2]Sun, Ke, et al. "Deep high-resolution representation learning for human pose estimation." arXiv preprint arXiv:1902.09212 (2019).
> [3]Chen, Yunpeng, et al. "Drop an octave: Reducing spatial redundancy in convolutional neural networks with octave convolution." arXiv preprint arXiv:1904.05049 (2019).
> [4]Chin, Ting-Wu, Ruizhou Ding, and Diana Marculescu. "Adascale: Towards real-time video object detection using adaptive scaling." arXiv preprint arXiv:1902.02910 (2019).
> [5]Ma, Ningning, et al. "Shufflenet v2: Practical guidelines for efficient cnn architecture design." Proceedings of the European Conference on Computer Vision (ECCV). 2018.
> [6]Yu, Jiahui, and Thomas Huang. "Universally slimmable networks and improved training techniques." arXiv preprint arXiv:1903.05134 (2019).

---

### Author Response · Authors · 2019-11-13
**General response to reviewers**

We sincerely thank all reviewers for their comments. We conclude our revisions as follows.

1. We add the training details and explanations in Appendix 1.
2. We add quantitative results to show the contribution of KL divergence in Appendix 2.
3. We reattach larger tables in Appendix 5 for better readability.

We would like to emphasize that our work is the first to bridge the gap between network structure and network input in a unified framework. The proposed method is general and effective, achieving competitive improvements on different structures, datasets and tasks. It is a plug-and-play strategy and easy to implement and reproduce. We also have released our codes for training and testing.

---

### Decision · Program_Chairs · 2019-12-19

**Decision:**

Reject

**Comment:**

Main content:new training regime for multi-resolution slimmable networks.

Discussion:
reviewer 4: believes the main contribution of mutual learning from width and resolution is a bit weak
reviewer 1: incremental work, details/baselines missing in experimental section
reviewer 2: (least detailed): well-written with good results
Recommendation: I agree with reviewer 1, 4 that the experimental section could be improved. Leaning to reject.